# Microeukaryotic Communities on the Fruit of *Gardenia thunbergia* Thunb. with a Focus on Pathogenic Fungi

**DOI:** 10.3390/pathogens10050555

**Published:** 2021-05-04

**Authors:** Bastian Steudel, Himansu Baijnath, Thorben Schwedt, Armin Otto Schmitt

**Affiliations:** 1Health and Environmental Sciences, Xi’an Jiaotong-Liverpool University, Suzhou 215123, China; bastiansteudel@aol.com; 2Ward Herbarium, School of Life Sciences, University of KwaZulu-Natal, Durban 4000, Kwa-Zulu Natal, South Africa; baijnathh@ukzn.ac.za; 3DLR German Aerospace Center, Bunsenstraße 10, 37073 Göttingen, Germany; thorben.schwedt@gmail.com; 4Institute of Maritime Energy Systems, DLR German Aerospace Center, Max-Planck-Straße 2, 21502 Geesthacht, Germany; 5Breeding Informatics, Department of Animal Sciences, Georg-August University of Göttingen, Margarethe von Wrangell-Weg 7, 37075 Göttingen, Germany; 6Center of Integrative Breeding Research, Albrecht Thaer-Weg 3, 37075 Göttingen, Germany

**Keywords:** environmental DNA, ITS2, global trading, ornamental plants

## Abstract

Woody fruit which stay on ornamental plants for a long time may present a risk of infection to other organisms due to the presence of pathogens on their surface. We compared the microbe communities on the fruit surfaces of garden ornamental *Gardenia* *thunbergia* Thunb. with those on other surfaces in the study region. As *Gardenia* fruit contain antifungal substances, the focus of this study was on the fungal communities that exist thereon. We used Illumina sequencing to identify Amplicon Sequence Variants (ASV) of the internal transcribed spacer 2 (ITS2) of the ribosomal RNA. The microbial communities of the *Gardenia* fruit are distinct from the communities from the surrounding environments, indicating a specialized microhabitat. We employed clustering methods to position unidentified ASVs relative to known ASVs. We identified a total of 56 ASVs representing high risk fungal species as putative plant pathogens exclusively found on the fruit of *Gardenia*. Additionally, we found several ASVs representing putative animal or human pathogens. Those pathogens were distributed over distinct fungi clades. The infection risk of the high diversity of putative pathogens represented on the *Gardenia* fruit needs to be elucidated in further investigations.

## 1. Introduction

Due to human activities, organisms have been moved around the globe over the last few centuries. Such translocations have led to invasions and naturalizations of alien species on all continents [1,2]. Some of these transports have been unintended, e.g., pathogens on crops, microbes on boots, crabs in water reservoirs of ships, etc. However, the introduction of hazardous organisms to other environments is a major threat to ecosystems and the agricultural and forest economy [3,4,5]. Recent studies revealed that microbial organisms are not “everywhere,” as previously stated as a paradigm [6,7]. Hence, studies on the microbial communities brought into new environments can be a possible measure of plant and animal/human disease prevention. In this study, methods of detection and strategies for preventing infections of plant pathogens are discussed intensively [8,9,10]. Although this study was initially focused on fungal plant pathogens as a source of threats to ecosystems as well as crop and ornamental plants in new environments, we identified several animal and human pathogens as well. Prominent taxa hazardous to both plants and animals are fungi of the Ascomycetes and Basidiomycetes. This study revealed different DNA sequences from those taxa to be present only on the fruit of *Gardenia thunbergia* Thunb. (Rubiaceae). To estimate the hazardous potential of transplanting *G. thunbergia* to other environments, molecular surveys of the ITS2 region were used. The ITS2 region is a very variable region between the 5.8S and the 28S ribosomal genes of eukaryotes. ITS2 is appropriate to determine taxa, especially fungi on the species level, if reference sequences are available [11,12].

Several fungi are well known as plant pathogens. However, different genera include pathogenic and nonpathogenic species [13,14,15], which can be difficult to distinguish. Further, host specificity varies substantially between pathogenic fungi [16]. On the other hand, the identification of host plants of pathogens is sometimes difficult. For example, it was shown that ornamental plants like orchids [15,17,18,19,20], agricultural plants [21], and woody plants [22,23] can be infected by *Fusarium* ssp., causing molds and leaf spots. Interestingly, *Fusarium* ssp. and *Acremonium* ssp. are among the taxa causing serious infections in humans [24,25]. Often, human or animal pathogens are of other taxa than plant pathogens, even if they may be sister clades to each other. Prominent animal pathogens are members of the Hypocreomycetidae like *Beauveria*, while other genera of this subclass like *Verticillium* are plant pathogens. *Fusarium* and *Acremonium* also belong to this subclass of Sordariomycetes in the Ascomycota. However, it is known that newly introduced plant pathogens, especially fungi, can be traced back to agricultural and ornamental plant imports and nurseries [26,27,28].

*G. thunbergia*, commonly called the white or forest gardenia, is an evergreen shrub or small tree usually growing up to five meters in height with strongly scented white flowers [29]. Its hard woody indehiscent fruit is consumed by large mammals, thus distributing the seeds. The species occurs naturally in South Africa (Eastern Cape and KwaZulu-Natal Provinces), Southern Mozambique, and eastern eSwatini (Swaziland). It was observed that the fruit of *G.thunbergia* persists for several years on the tree and may therefore host pathogens. Hence, the fruit of this species was studied. It was reported that different pathogens were introduced to Egypt through *Gardenia jasminoides* J.Ellis [30]. We expected adapted fungi communities in the samples of *G.thunbergia*, as it was previously shown that the fruit of other *Gardenia* species bear antifungal ingredients [31,32], although contaminations with fungi on the dried fruit of *G. jasminoides*, which is used as medical drug, are known [33].

## 2. Results

### 2.1. General Overview on the Microeukaryote Communities of All Samples

*G. thunbergia* fruit samples (Figure 1) were found to be clustered closely together (Figure 2).

The samples from the whole study area are more widespread than the *Gardenia* fruit samples, while sample Afr_4 [*Syagrus romanzoffiana* (Cham.) Glassman] is placed in the lower-right part of the graphs, next to the *G*. *thunbergia* samples. 

Including the nonidentified ASVs under study, 58 Alveolata, 3243 fungi, 3 Heterobolosa, 32 Metazoa, 44 Protista, and 539 microalgae were found, making a total of 3919 ASVs. Most of the Alveolata, Heterobolosa, and Metazoa were identified at the genus level, whereas the Protista were not been identified at a lower taxon level. However, this study focused on the ASV of fungi for further analyses to identify putative pathogens.

A comparison between the sequences found on the fruit of *G. thunbergia* (samples Afr_10 to Afr_16) and the other samples (Afr_1 to Afr_9) showed that 1029 sequences of the fungi (33%) were found in both groups, while 261 (8%) were only found on the *Gardenia* fruit. In contrast, 172 sequences of microalgae (32%) were present in both groups, but only 1 (0.2%) was found exclusively on the fruit. Thus, the fungi found exclusively on the fruit of *Gardenia* do not stem from lichen of the surrounding environment. Hence, they may represent lichen fungi which are infecting the same microalgae as other fungi in the study region, or do not stem from lichen growing on the fruit. If those fungi are not endosymbionts of lichen, they may represent pathogens.

### 2.2. Abundances of the Microbial Communities

In total, 2.57 million reads were recorded, of which 1.06 million were from *Gardenia thunbergia* Thunb. fruit. The most common organisms were fungi and lichen algae. Out of the records with more than 1% of total reads, 12 fungi and 2 microalgae were found; the latter had the lowest read numbers. Focusing on the *Gardenia* fruit, we identified 62,541 reads from organisms that were present only on the fruit and not in the other local communities. After removal of ASVs from other organisms, 261 fungus ASVs were recorded by 62,337 reads. The most abundant sequence was ASV_000076 (*Candida homilentona*, 11,153 reads), followed by ASV_000487 (*Penicillium* sp., 3204 reads), and ASV_003838 (*Cosmospora* sp., 2445 reads). 113 (43%) ASVs which were found only on the *Gardenia* fruit accounted for more than 0.01% of the reads from the *Gardenia* samples, while 27 (48%) of the 56 high risk ASVs accounted for more than 0.01% of the reads from the *Gardenia* fruit. 

### 2.3. Diversity and Composition of Fungi on Fruit of Gardenia Thunbergia

The dendrogram including all fungus sequences exclusively from the *Gardenia* fruit (Appendix A) showed a cluster of Basidiomycota (cluster A), four clusters of Ascomycota (clusters B to E), and a nonphylogenetic group of Ascomycota (F), which was merged into one clustering as otherwise three very small groups would have been necessary (Appendix A). A color code was used to identify high risk clades for plant and animal pathogens (red for plant, and brown for human/animal pathogens) in the six single group dendrograms. Clades that most probably bear only nonpathogenic species are marked in dark green, while those that are unlikely to bear pathogens are marked in bright green. Yellow marked clades cannot be estimated regarding their pathogen potential. A combination of the respective dendrogram letter and a consecutive number to identify clades or sequences that were not identified to genus level was used.

The dendrogram of cluster A (Basidiomycota) revealed five taxa (Appendix A). Most of the sequences belonged to the Tremellomycetes, forming three (or four if A1 is in the Tremellomycetes) taxa (*Kockovaella*/*Fellomyces*, A2, and *Tremella*), of which one comprised the genus *Septobasidium* (Pucciniomycetes). The A1 clade may be any class of Basidiomycota, as it stands between *Septobasidium* and the Tremellomycetes clade.

The dendrogram of cluster B (Appendix A) shows four groups and two single sequences that cannot be determined at the genus level or are unknown. Clade B1 comprises two sequences of Lecanoromycetes. Clade B2 contains Didymosphaeriaceae, clade B3 is not further classified, and the sequence B4 is sister to Funbolia. Clade B5 is sister to sequence B6 and the *Xanthomendoza weberi* reference sequence. Further, sequence ASV_004060 was identified as *Diplodia* and was placed next to the pathogen *Macrophomina phaseolina*. Hence, this sequence may represent a possible plant pathogen fungus. The pathogens *Thielaviopsis* and *Ceratocystidiaceae* sp. “MF952419” are clustered with the lichen genus *Xylaria*, all other reference pathogen sequences included in cluster B are the sister group of these.

Cluster C shows pathogen taxa from the Hypocreales (Figure 3).

One main clade is built by the animal and human pathogen genera *Cyphellophora*, *Exophilia*, and *Cladophialophora*, including the C1 sequence ASV_003637. The clade of the genus *Cyphellophora* is monophyletic and includes the previously unidentified sequence ASV_000635. The clade of the genus *Exophilia* is monophyletic and includes three sequences (ASV_001177, ASV_000886, and ASV_004028) which could not be identified.

The other main clade is formed by the plant pathogens, including all plant pathogen references of this clade, the genera *Eutype* and *Sarcocladium*, and the genera *Diaporthe* and *Phomopsis* from the Diaporthales. Clades C2 (Pleosporaceae) and C3 (Nectriaceae) are nested in this clade. However, these clades include single sequences determined to genus level. The sequences ASV_003826 and ASV_001699 of C2 belong to the Pleosporaceae, while the sequence ASV_003068 was identified as *Neooccultibambus*, and the sequence ASV_002435 as *Crassiparies*, both from the Pleosporaceae. The remaining three sequences of this clade were not identified further. ASV_004044 belongs to *Pectenia* (Peltigerales), and is the basis to the lichen reference sequences of *Cladonia* and sequence C4, while *Sarcocladium* and the Nectriaceae are nested into this clade.

Cluster D (Appendix A) includes mainly *Penicillium* and *Talaromyces* sequences as a sister group to lichen fungi (Pezizomycotina). Talaromyces A clade is the basis of D1, Penicillium A, and the Talaromyces B clade. Talaromyces C sequence ASV_001465 (*Talaromyces verriculosus*) is the sister to the two Hypocreales sequences D4 (ASV_002546 and ASV_003642). The basis of the *Penicillium* cluster is built by the D5 (Eurotiomycetes) clade. D6 is included in the Penicilium B and Penicillium C cluster and most likely represents a *Penicillium* species. D7 is the sister group to Penicillium D, including *Xylographa trunciseda*, while *X. trunciseda* seems to be misplaced.

In cluster E (Figure 4), only Sordariomycetes, with only four sequences from *Phaeocremonium* (Togniniales) split into two branches (ASV_003665, ASV_000840, and ASV_000997) and ASV_004056, and all other classified sequences from the Hypocreales were identified.

However, some of the sequences were not identified deeper than to the level of Sordariomycetes, and may hence represent other orders. Overall, the three genera *Acremonium* (three clades), *Phaeocremonium* (two clades), and *Niesslia* (two clades) are split, while the reference sequence pairs of *Verticillium* and *Fusarium* are distinct. The genera *Beauveria*, *Sarcocladium*, *Trichothecium*, *Clonostachys*, *Pseudocosmophora*, and *Rosaspaeria* each form monophyletic clades. The E5 (Nectriaceae A) clade is the basis of E6 (Nectriaceae), the genus *Pseudocosmophora* (Nectriaceae), E7, E8, and *Verticillium leptobracteatum*. Clade E10 bears the sequence ASV_004048 (*Rosasphaeria*), E11 cannot be placed, although it is sister to *Microcera*.

Cluster F (Appendix A) was built by the sequences not included in the clusters A–E, as it would be inappropriate to build three very small clusters of those remaining according to their position in the overall phylogeny. F1 is the sister to F2 (Eurotiales), the genus *Arachnomyces* is the sister clade to F3, *Candida*, and the reference sequences of *Taphrina*.

Sequences representing organisms with a high risk of being plant pathogens with sequence abundances and fruit age are shown in Table 1.

The abundances of sequence numbers of the 56 high risk taxa in our sampling are very different, ranging from 10 to 2445.

We recorded an estimated age gradient of the fruit of *G. thunbergia* under study from sample Afr_10 (youngest), Afr_16, Afr_13, Afr_12, Afr_14, Afr_15, to Afr_11 (oldest) based on the density and size of the on-growing lichens, but the last three samples (Afr_14, Afr_15, and Afr_11) were very similar regarding their lichen growth, and thus, may be similar in age. We classified the samples Afr_10 and Afr_16 as young, the samples Afr_13 and Afr_12 as medium age, and the samples Afr_14, Afr_15, and Afr_11 as old based on the coverage and size of the on-grown lichen. We found 27 of the high risk sequences on the young, 24 on the medium aged, and 35 on the old fruit.

## 3. Discussion

### 3.1. Overall Microeukaryote Communities

Microeukaryote communities from the fruit of *G. thunbergia* (samples Afr_10-Afr_16) are different from those of other host plants and artificial surfaces (samples Afr_1-Afr_9) from our study region. For these special habitats, a more detailed analysis is required of the putative pathogens of these communities. These distinct microbial communities of *G. thunbergia* fruit were expected, as the genus is known to bear antifungal substances. However, it is remarkable that we found only one microalga on the fruit, but 261 fungi ASVs. For the other microalgae and fungi, more than 30% of the found ASVs were recorded on the *Gardenia* fruit and the other samples, while the rest was only found in the non-*Gardenia* samples. While we expected to observe only a part of the microbial community of a region on one specialized habitat (*Gardenia* fruit), this clearly shows that the lichen on *G. thunbergia* fruit bear photobionts that are not specialized to this habitat. In fact, we found only very few microfungi considered as lichen forming on the *Gardenia* fruit.

### 3.2. Pathogen Fungi on Gardenia thunbergia Fruit

A great variety of fungus sequences representing taxa of possible plant and animal pathogens was only present on *G. thunbergia* fruit. These are from the Basidiomycetes and the Ascomycetes. Generally, estimates of the hazardous potential of different species of a genus can be difficult, and morphological determination can mismatch the underlying phylogeny [13,34]. Therefore, we identified clades that may be hazardous if the genus is known to bear mainly pathogenic species or is nested into hazardous species/genera in cases of species that are unknown to genus level.

In the Basiodiomycetes group of our sequences, we found only non- or likely nonhazardous clades. *Septobasidium* is known to form symbiotic associations with insects, but is not associated with plant parasitism. The remaining clades are unlikely to be hazardous to plants due to their position in the dendrogram.

In the Ascomycetes groups of the fungi occurring only on the *Gardenia* fruit, clusters were identified with high risk of plant (and animal) pathogens and others with low risk of containing plant pathogens. While clusters C and E include the most plant pathogens, cluster F represents numerous animal pathogens. In cluster C, the genera *Xyphellophora*, *Exophilia*, and *Cladophialophora* are known as animal and human pathogens or infectious. The other half of the dendrogram bears the included pathogen reference sequences and needs to be considered as plant pathogen taxa, except the sequence ASV_004044 (Pectenia) and ASV_002615 (unidentified). Note that the reference sequences of *Cladonia* are included there. Cluster E includes known animal pathogens (*Acromonium* on mammals and *Beauveria* on Arthropods) and known plant pathogens, while smaller groups cannot be identified as plant pathogens but may nonetheless be pathogenic.

Clusters B and D may include taxa representing plant pathogens, but this is difficult to estimate as the clade B1 is nested in plant pathogen reference sequences, and *Xylaria*, which is not known as plant pathogenic but rather a saprophytic genus, is the crown group of the pathogen reference sequences. On the other hand, the B3 and B5 clades are related to lichen reference sequences. In cluster D the closely related genera *Penicillium* and *Talaromyces* form the major group in the dendrogram. Both genera inhabit pathogens and nonpathogenic species. Hence, it cannot be judged whether they represent a possible threat of infection in foreign habitats.

Focusing on the number of likely pathogenic fungi taxa, we identified 18 sequences in the clade C, and 38 sequences in the clade E as „high risk“ taxa for plant pathogens (total 56). Of these, 13 (clade C) and 25 (clade E), respectively, were not determined to genus level (Table 1). Hence, these taxa may represent thus far unknown pathogens. The human and animal pathogen sequences could be determined on genus level, however.

The abundance of the ASVs representing high risk organisms is very different. It may be speculated that the infection risk depends on the number of organisms in the different samples. However, as we used Illumina sequencing with a prior PCR, the number of sequences detected may not represent the corresponding abundance or relations in the living biofilm. Irrespective of that, about half of the high risk ASVs were found with a read abundance of more than 106 counts (0.01% of the reads from *Gardenia* fruit), indicating that these are present at a certain quantity in the biofilms studied. As the number of ASVs is highest (35 out of 56) on the oldest fruit, the infectious potential of old *G. thunbergia* fruit may be considered higher than that of young or medium-aged fruit.

### 3.3. Lichen on Gardenia thunbergia Fruit

As the old fruit of *G. thunbergia* that were studied were covered with thick lichen biofilms or crusts, we expected some extraordinary lichens as well. However, exclusively on the fruit of *G. thunbergia*, only three sequences (ASV_000986, ASV_001217, and ASV_004044) were found that were identified as lichen fungi (Pezizomycetes), and two additional sequences (ASV_004052 and ASV_002615) that may represent a lichen fungus. This, however, corresponds to the fact that only a single microalga was found exclusively on the fruit. Hence, lichen growing on the fruit of *G. thunbergia* are most likely not restricted to this habitat. This is somewhat surprising, as it we expected to observe lichen and fungi on the fruit surfaces, which are adapted to the fruit ingredients of *G. thunbergia*, since the fruit of *G. jasminoides* contains cerbinal, an antimycobial ingredient [31]. Further, it was reported that also *G. brighamii* H.Mann leaf extracts contain antifungal substances [32]. On the other hand, many fungi taxa found exclusively on the fruit of *G. thunbergia*, which may be a threat to other environments, crops and garden ornamentals, were identified.

## 4. Materials and Methods

### 4.1. Methods Overview

We analyzed 16 samples of microbial mats or biofilms from different plant and artificial surfaces in the Durban region in South Africa (Figure 1). We used nine samples from different habitats to determine the general microbial communities at the study site (samples Afr_1 to Afr_9) and compared them with the samples of *Gardenia* fruit (samples Afr_10 to Afr_16). Three replicates were taken from each sample and processed individually, resulting in a total of 48 DNA extractions. For identification of the microbial communities, we used Illumina sequencing to produce libraries of ITS2 sequences of all eukaryotes in these samples. After identification of sequences that were unique to the fruit surfaces of *G. thunbergia* possible pathogenic taxa were identified.

### 4.2. Study Region and Sample Collection

Samples were collected in “Strawberry Fields” in KwaZulu-Natal, Durban, Sherwood, RSA, and nearby. Sherwood is known as the “garden suburb“ of Durban, South Africa. Durban is characterized by a warm temperate climate with an annual mean temperature of almost 21 °C and a mean annual precipitation of about 1000 mm. From May to October, temperatures are lower than the year mean temperature, precipitation is lower from April to September than in the other months.

The seven samples of *G. thunbergia* fruit (samples Afr_10 to Afr_16) were obtained exclusively from Strawberry Fields, while the others were collected at Strawberry Fields (samples Afr_1, Afr_4, Afr_5, Afr_6, and Afr_9) and other sites (samples Afr_2, Afr_3, Afr_7, and Afr_8). Non-*Gardenia* samples were taken from different surfaces like concrete, tree barks, and a leaf scale. The *Gardenia* fruit was collected whole, while the hard surface samples were scratched with a scalpel and the plant surface samples were cut from the plant. An overview on the samples is given in Appendix A. All samples were dried indoor at room temperature on separate paper sheets to avoid contamination in Durban, put into plastic jars for shipment, and stored after shipping in a freezer at −20 °C until sample processing.

### 4.3. DNA Preparation and PCR

Three replications of DNA extraction per sample were obtained using surface material from different spots of the sample surface by scalpel scratching. This material included the surface material of the sample, i.e., epidermis and some of the tissue below the epidermis of the plant samples. This was done to include organisms growing on the surface and within the plant material. After removal, the material of each replicate was ground separately in a mortar without liquid nitrogen to obtain an even sample material; later, it was further ground during the DNA extraction procedure. The DNeasy PowerSoil Pro Kit (Qiagen, Hilden, Germany) was used as specified in the manual for DNA extraction. After DNA extraction, a PCR with the primers ITS3_KYO2 (GATGAAGAACGYAGYRAA) ([35]) and ITS4 (TCCTCCGCTTATTGATATGC) ([36]) was done, including the default MiSeq adapters. The volume of the PCR reactions was 50 µL, containing 1µL of each primer solution (10 µM), 10 µL 5× GC Buffer (Thermo Scientific, Waltham, MA, USA), 0.2 µL MgCl_2_ (25 mM, Thermo Scientific, Waltham, MA, USA), 2.5 µL DMSO (5%, Thermo Scientific, Waltham, MA, USA), 1 µL dNTPs (10 mM, Thermo Scientific, Waltham, MA, USA), 0.5 µL Phusion High-Fidelity DNA polymerase (10 mM, Thermo Scientific, Waltham, MA, USA), 1 µL template (containing 25 ng DNA), and 32.8 H_2_O. Denaturation was done at 98 °C for 1 min, 25 cycles were performed at 98 °C for 45 s, 60 °C for 45 s, and 72 °C for 30 s, and a final extension was done at 72 °C for 5 min. Negative controls were prepared without template but with 1 µL H_2_O, and positive controls were prepared with a template known to bear DNA from a previous study. PCR products were concentrated and purified with MagSi-NGS^Prep^ magnetic beads, as recommended by the manufacturer (Steinbrenner, Wiesenbach, Germany). DNA was eluted in 30 µL elution buffer EB (Qiagen, Hilden, Germany). Purified PCR products were quantified and sequenced as described by Schneider and colleagues [37] using a MiSeq instrument and v3 chemistry (Illumina, San Diego, CA, USA).

### 4.4. Data Processing and Selection of Sequences

The replicates Afr_3a, Afr_3b, and Afr_15c could not be amplified successfully, and were therefore excluded from further analyses. From the other runs, we obtained on average 57,277 sequence reads per sample replicate, with a minimum of 11,382 and a maximum of 128,359.

Paired-end sequencing data from the Illumina MiSeq were processed as follows. Sequences were quality-filtered with fastp (version 0.19.6) ([38]) using a per base phred score of 20, base pair correction by overlap (option -c), as well as 5′- and 3′-end read trimming with a sliding window of 4, a mean quality of 20 and minimum sequence size of 50 bp. After quality control, the paired-end reads were merged using PEAR (version 0.9.11) ([39]), primers were clipped using cutadapt (version 1.18) ([40]) with default settings and further processed using VSEARCH (v 2.9.1) ([41]). This included size sorting and filtering of the paired reads to a minimum of 140 bp (--sortbylength --minseqlength 140) and dereplication (--derep_fulllength). Dereplicated ASVs were denoised with UNOISE3 using default settings (--cluster_unoise – minsize 8) and chimeras were removed (--uchime3_denovo). Additional reference-based chimera removal was performed (--uchime_ref) against the SILVA SSU NR database (https://www.arb-silva.de/, accessed on 6 January 2020). A total of 4290 ASVs were identified in the samples. Taxonomic assignment was performed against the UNITE database v8.2 (https://plutof.ut.ee/#/doi/10.15156/BIO/786372, accessed on 6 January 2020) with an identity of at least 90% to the query sequence resulting in a total of 3437 ASVs, of which 3146 have been identified as “fungi” and 291 had no blast hits. As most microalgae were excluded by the processing, probably because they may not be included in the data base, we reconsidered those excluded ASVs that had a minimum of 80 sequence reads in at least two different sample replicates. The resulting “no blast hit“ sequences were manually grouped into fungi or Viridiplantae using the NCBI database (https://www.ncbi.nlm.nih.gov/, accessed on 7–12 September 2020) for inclusion into further analyses. This additional threshold to omit artifacts was used as it is very unlikely to record 80 sequence reads resulting from chimera or other artifacts [42]. Only for a single sequence there was no similarity found in the NCBI data base with standard settings. This resulted in a total of 4067 different ASVs. We removed 148 sequences from Pteridophyta, Bryophyta, and Anthophyta (e.g., *Euphorbia*, *Fallopia*, *Gardenia*, *Libidibia*, and *Yucca*) to keep only sequences of the microalgae as photosymbiontic organisms, resulting in a total of 3919 sequences which we included in the analyses.

ASVs instead of operational taxonomic units (OTU) were used for our analyses as ASV can resolve the biological diversity more exactly [42,43]. 

For the description of the sequence abundances, the mean values of the sequence numbers of the three replicates for all samples was used. This was chosen as generally very strong correlation between the replicates of each sample (Appendix A), with a mean correlation coefficient of almost 0.9 was found.

### 4.5. Data Records

The data ITS2 amplicon raw sequences were deposited at the National Center for Biotechnology Information under the BioProject PRJNA672041.

### 4.6. Statistics

Principal Component Analyses (PCA) were performed using the free machine learning library “scikit-learn” for the Python programming language. Data were provided in CSV format and read using Pandas (pandas.read_csv—pandas 1.2.3 documentation (pydata.org)) with 45 rows representing the different replicates and 4290 columns, representing the ASVs. The dataset was normalized using the scikit-learn StandardScaler (sklearn.preprocessing.StandardScaler—scikit-learn 0.24.1 documentation (scikit-learn.org)) to a mean of 0 and standard deviation 1. The first 10 principal components were calculated using sklearn (sklearn.decomposition.PCA—scikit-learn 0.24.1 documentation (scikit-learn.org) ([44]) and “fit_transform(x)”, where x is the normalized dataset and the explained variance was available with the attribute (“explained_variance_ratio”). Results were plotted using matplotlib (Matplotlib: Python plotting—Matplotlib 3.3.4 documentation). The same procedure was performed for the presence/absence data set in which the read abundances were omitted. 

### 4.7. Construction of Dendrograms

Focusing on the 261 sequences of fungi exclusively found on the *Gardenia* fruit, we constructed a dendrogram including known pathogens and lichen fungi as references. This was necessary for sequence assembly for the ASV that were not identified during the previous data processing (e.g., “uncultured fungi”, “no blast hit”). For this, distance matrices produced by using the package “kmer” ([45]) in R 4.0.2 ([46]) were built. The six groups identified in this dendrogram were used to construct individual dendrograms accordingly. Eight reference sequences (two *Phytophtora*, two *Pythium*, two *Lichenomphalia*, and two *Colletotrichum*) were omitted in the separate analyses of the six groups as no related taxa were found (Oomycota, Agaricomycetes, and Glomerellales, respectively) in the ASVs exclusively from the samples from the *Gardenia* fruit.

## 5. Conclusions

The fruit of *G. thunbergia* bears a great variety of pathogens to plants as well as to humans and animals. The pathogens found on the fruit of *G. thunbergia* may pose a threat to plants and animals, especially if they are brought into environments to which they have not yet adapted. How the infection risk of these microbial communities needs to be considered regarding phytopathology measures requires further study. 

## Figures and Tables

**Figure 1 pathogens-10-00555-f001:**
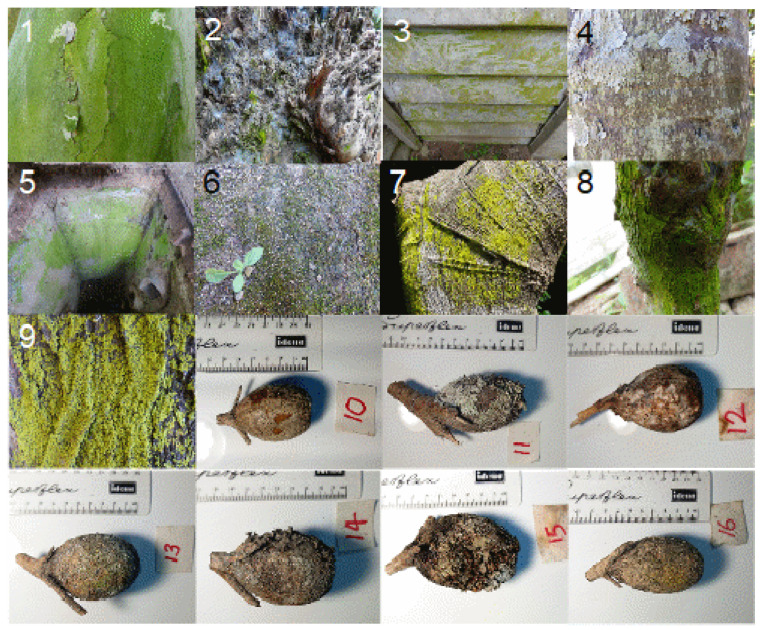
Samples from South Africa used in this study. Samples Afr_1toAfr_9 (pictures 1 to 9) are from different surfaces of plants and artificial substrates and are used to characterize the microbial communities of the whole study site, while samples Afr_10 to Afr_16 are *Gardenia thunbergia* fruit (see Appendix A).

**Figure 2 pathogens-10-00555-f002:**
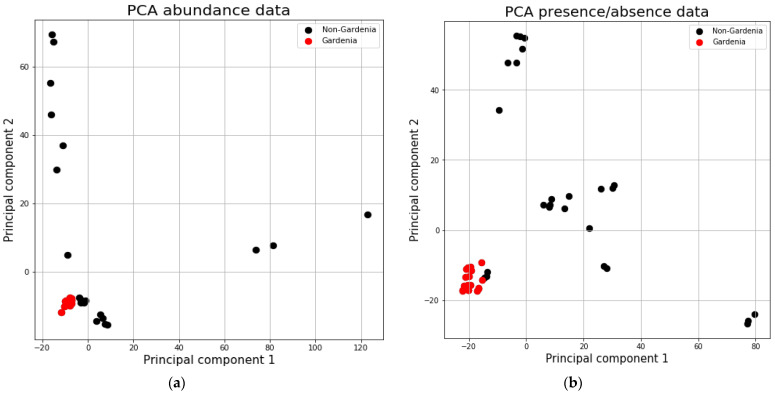
Principal Component Analyses (PCA) of the microbial communities of all samples with abundance data (**a**) and presence/absence data (**b**). Note that Gardenia fruit samples are clustered on the down-left side (red dots) while the other samples (black dots) are widely distributed for both analyses. Principal Component 1 has an explanatory power of 15.8% and 16.5%, and Principal Component 2 of 10.7% and 13.2% for the abundance and the presence/absence data, respectively.

**Figure 3 pathogens-10-00555-f003:**
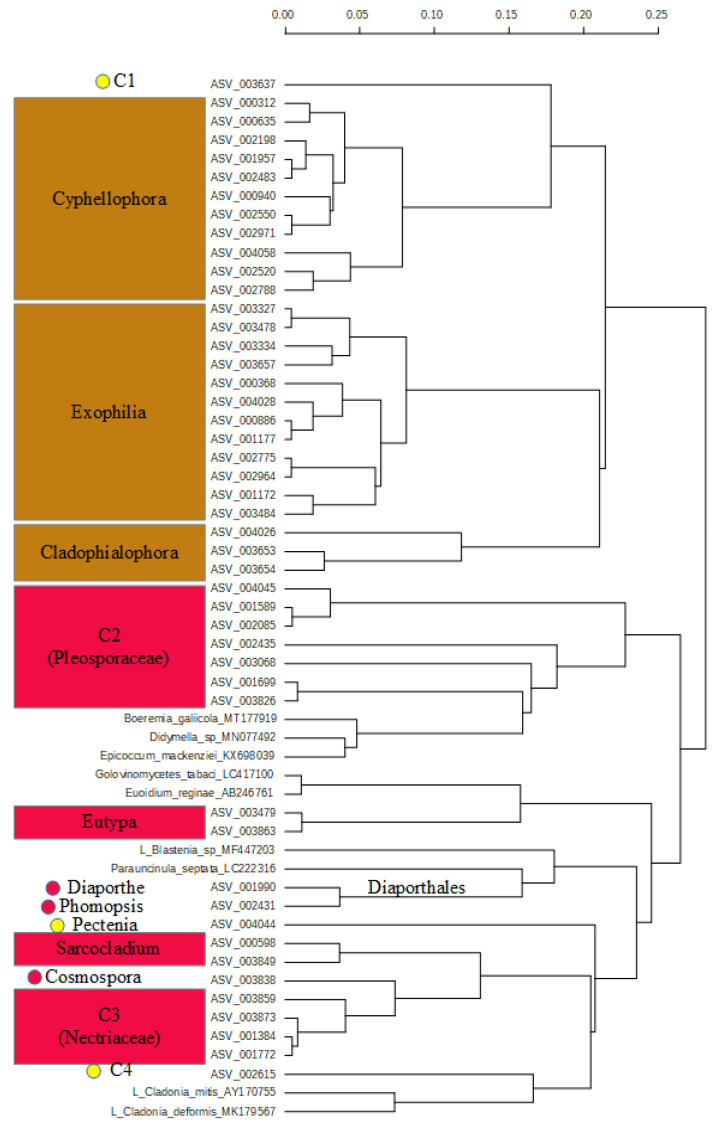
Dendrogram of cluster C. Almost all sequences belong to pathogens. Brown clusters represent pathogens to animals and humans with only few plant pathogens, red clusters are likely plant pathogens. Dots are used for singleton sequences. Due to space limitation “Diaporthales“ was written on the branch of the cluster of *Diaporthe* and *Phomopsis.* The level of difference is given in the scale.

**Figure 4 pathogens-10-00555-f004:**
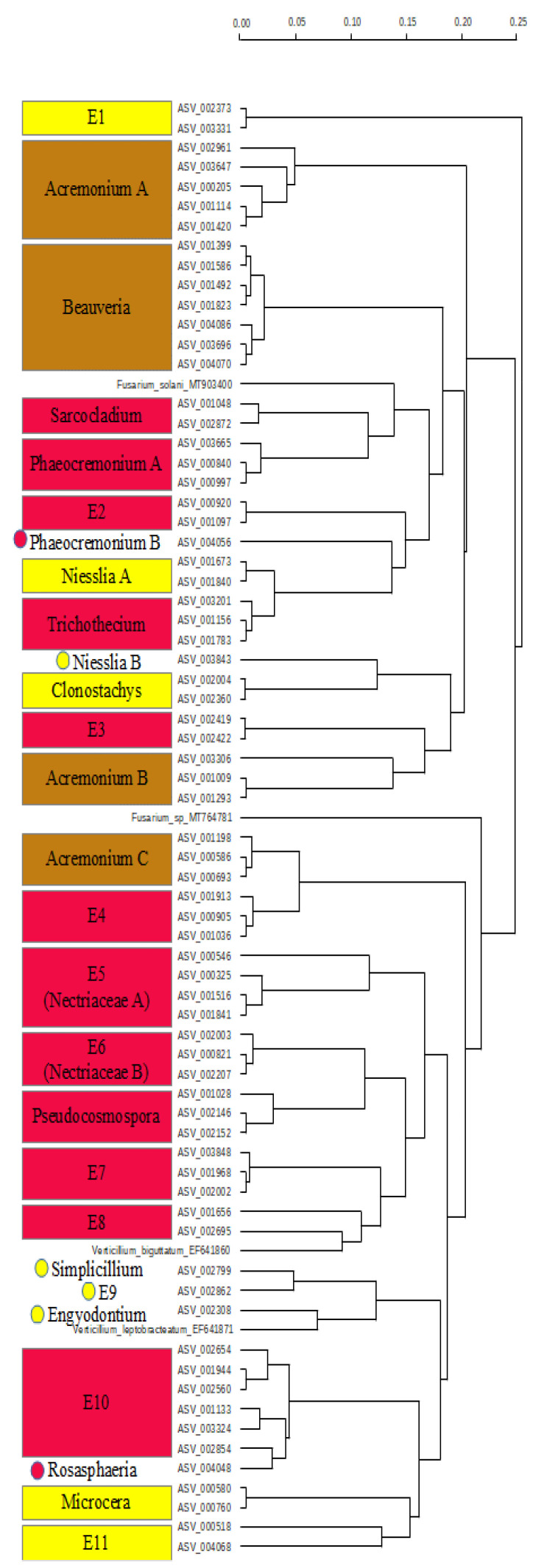
Dendrogram of cluster E. Yellow clades cannot be classified as plant pathogens, brown clades are animal pathogens, and red clades represent most likely plant pathogens. Dots are used for singleton sequences. The level of difference is given in the scale.

**Table 1 pathogens-10-00555-t001:** Overview of the sequences rated as representing high risk plant pathogens. Sequence ID stands for sequence identity, no. seq stands for total number of sequences, no. samples stands for number of samples in which the sequence was found, the age was classified as “young” (y), “medium” (m), and “old” (o). If the respective sequence was found in all age classes, this was noted as “all”. If the family or the genus of the organism represented by the ASV in not determined this is indicated by a “?”.

Sequence ID	Family	Genus	Cluster	No. seq.	No. samp.	Age
ASV_004045	Pleosporaceae	?	C	21	1	y
ASV_001589	Pleosporaceae	?	C	94	2	y
ASV_002085	Pleosporaceae	?	C	55	1	y
ASV_002435	Pleosporaceae	?	C	133	5	all
ASV_003068	Pleosporaceae	?	C	33	1	o
ASV_001699	Pleosporaceae	?	C	57	1	y
ASV_003826	Pleosporaceae	?	C	10	1	y
ASV_003479	Diatrypaceae	Eutypa	C	31	1	o
ASV_003863	Diatrypaceae	Eutypa	C	18	1	o
ASV_001990	?	?	C	365	1	m
ASV_002431	?	?	C	309	1	m
ASV_000598	incertae sedis	Sarcocladium	C	737	3	y/o
ASV_003849	incertae sedis	Sarcocladium	C	26	1	y
ASV_003838	Nectriaceae	Cosmospora	C	2445	7	all
ASV_003859	Nectriaceae	?	C	160	1	o
ASV_003873	Nectriaceae	?	C	23	1	o
ASV_001384	Nectriaceae	?	C	176	1	o
ASV_001772	Nectriaceae	?	C	144	1	o
ASV_001048	incertae sedis	Sarcocladium	E	435	1	o
ASV_002872	incertae sedis	Sarcocladium	E	51	1	o
ASV_003665	Togniniaceae	Phaeocremonium	E	66	1	o
ASV_000840	Togniniaceae	Phaeocremonium	E	240	1	o
ASV_000997	Togniniaceae	Phaeocremonium	E	168	1	o
ASV_000920	?	?	E	184	1	y
ASV_001097	?	?	E	148	1	y
ASV_004056	Togniniaceae	Phaeocremonium	E	48	1	m
ASV_003201	incertae sedis	Trichothecium	E	28	2	y/m
ASV_001156	incertae sedis	Trichothecium	E	146	2	y/m
ASV_001783	incertae sedis	Trichothecium	E	87	2	y/m
ASV_002419	?	?	E	39	3	y/o
ASV_002422	?	?	E	34	3	y/o
ASV_001913	?	?	E	147	2	y/o
ASV_000905	?	?	E	164	3	all
ASV_001036	?	?	E	190	4	y/o
ASV_000546	Nectriaceae	?	E	662	1	m
ASV_000325	Nectriaceae	?	E	769	1	o
ASV_001516	Nectriaceae	?	E	111	2	m/o
ASV_001841	Nectriaceae	?	E	83	1	o
ASV_002003	Nectriaceae	?	E	56	2	y/m
ASV_000821	Nectriaceae	?	E	236	1	m
ASV_002207	Nectriaceae	?	E	37	1	m
ASV_001028	Nectriaceae	Pseudocosmophora	E	512	1	m
ASV_002146	Nectriaceae	Pseudocosmophora	E	461	1	m
ASV_002152	Nectriaceae	Pseudocosmophora	E	189	1	m
ASV_003848	?	?	E	17	2	y/o
ASV_001968	?	?	E	60	1	o
ASV_002002	?	?	E	55	1	o
ASV_001656	?	?	E	88	2	o
ASV_002695	?	?	E	56	2	y/o
ASV_002654	?	?	E	178	4	all
ASV_001944	?	?	E	414	4	m/o
ASV_002560	?	?	E	128	4	all
ASV_001133	?	?	E	386	3	all
ASV_003324	?	?	E	469	5	all
ASV_002854	?	?	E	273	2	m/o
ASV_004048	Niessliaceae	Rosasphaeria	E	37	5	all

## Data Availability

The data ITS2 amplicon raw sequences were deposited at the National Center for Biotechnology Information under the BioProject PRJNA672041.

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
