# Peer review of "Microeukaryotic Communities on the Fruit of Gardenia thunbergia Thunb. with a Focus on Pathogenic Fungi"

_pathogens, 2021, doi:10.3390/pathogens10050555_

Round 1
Reviewer 1 Report
Article entitled “Fungi on the fruits of Gardenia thunbergia Thunb. with a focus on plant pathogens” aimed at studying the molecular surveys and variability of ITS 2 to assess the hazardous potential of cultivating G. thunbergia in another environment.
In this manuscript, the main idea is well covered, organized, and coherent. Language is good but many spelling mistakes and typing error in the whole manuscript. In general, after major corrections and clarifications,
Below will find observations and comments
- The title should be changed to match the main findings of the research work (Micro-eukaryotic community on the fruits of Gardenia thunbergia Thunb. with a focus on pathogenic fungi).
- Abstract is very poor and does not contain the main findings of the work.
- The introduction is good enough, but focused on plant pathogenic fungi, while the results covered many animal and human pathogenic fungi. Hence, the introduction should offer some information about the fungal groups that invade humans and animals
- Results: Line 79; ASV write it in the full term and put the abbreviation in parentheses
- Line 82; authors stated that “this study focused on the fungi and microalgae” while in introduction said in line 36 “this study focused on fungal plant pathogen” although in the abstract line 21 said “ we identify several clades of fungi with possible pathogenic species to animals ……” Therefore, the authors need to clarify the exact idea of the research work and stick to it throughout the entire manuscript.
- In line 86; authors refer to the fruits of thunbergia and other samples as two groups while in Materials and Methods either in Methods overview or in study region and sample collection authors did not divided the 16 samples into two different groups. So, it should separate in the experimental design first to become understandable in the results.
- In line 90; authors stated that “lichen fungi which are infecting the same microalgae” infecting is not a right word it is better to be inhabiting the same microalgae.
- If the journal regulation allows; it is better to merge the results with discussion as many findings should be supported by references such as in line 140; “animal and human pathogen genera Cyphellophora, Exophilia and …….” this sentence need reference confirming these genera are animal and human pathogen (many others are same). Consequently, I also recommended that authors design a table including plant, animal, and human pathogenic fungi with host and in the last column they can add the references.
Regarding Materials and methods,
- what is the sampling technique used in this work?
- Authors should write the GPS location of the study region and adding a satellite image for this area.
- Also, authors need to explain why he choose this region; is it the only region in south Africa that has G. thunbergia. If it is not, authors should mention about the distribution of this ornamented plant in the south Africa in Introduction part.
- Finally, I would like to mention that all tables and figures in supplementary data are not appeared except table 2
Best regards
Author Response
Author's Reply to the Review Report (Reviewer 1)
Dear reviewer, unfortunately, we overlooked one ASV of a putative pathogen (ASV_003838, Cosmospora) previously. Hence, we needed to adjust the numbers in the text accordingly and included this sequence in Table1. Further, some minor changes in the results section (lines 225-228) and in the discussion (338-341 and 379-385) were necessary, as this led to an other interpretation of another sequence (ASV_004044) as well (being considered as a lichen fungus). In the whole ms, we omitted some minor typos and removed double spaces. All these corrections are marked with track changes.
In this manuscript, the main idea is well covered, organized, and coherent. Language is good but many spelling mistakes and typing error in the whole manuscript. In general, after major corrections and clarifications,
Below will find observations and comments
The title should be changed to match the main findings of the research work (Micro-eukaryotic community on the fruits of Gardenia thunbergia Thunb. with a focus on pathogenic fungi).
- we changed the title accordingly
Abstract is very poor and does not contain the main findings of the work.
- we re-phrased parts of the abstract to present more details of our findings
The introduction is good enough, but focused on plant pathogenic fungi, while the results covered many animal and human pathogenic fungi. Hence, the introduction should offer some information about the fungal groups that invade humans and animals
- we included some details on human and animal pathogens
Results: Line 79; ASV write it in the full term and put the abbreviation in parentheses
- as we used ASV now in the abstract, we do not give the full term here again. Similarly, we included the term ITS2 in the abstract.
Line 82; authors stated that “this study focused on the fungi and microalgae” while in introduction said in line 36 “this study focused on fungal plant pathogen” although in the abstract line 21 said “ we identify several clades of fungi with possible pathogenic species to animals ……” Therefore, the authors need to clarify the exact idea of the research work and stick to it throughout the entire manuscript.
- we omitted the microalgae in this paragraph and added that we will focus later on the pathogens. This is reading now:
“However, this study focused on the ASV of fungi for further analyses to identify putative pathogens.“
In line 86; authors refer to the fruits of thunbergia and other samples as two groups while in Materials and Methods either in Methods overview or in study region and sample collection authors did not divided the 16 samples into two different groups. So, it should separate in the experimental design first to become understandable in the results.
-we now included the numbers of the samples in parentheses there and included the sentence
“We used nine samples from different habitats to determine the general microbial communities at the study site (samples Afr_1 to Afr_9) and compared them with the samples of the Gardenia fruits (Afr_10 to Afr_16).“
to the methods overview to determine the two study groups.
In line 90; authors stated that “lichen fungi which are infecting the same microalgae” infecting is not a right word it is better to be inhabiting the same microalgae.
- we toatally agree and rephrased this accordingly
If the journal regulation allows; it is better to merge the results with discussion as many findings should be supported by references such as in line 140; “animal and human pathogen genera Cyphellophora, Exophilia and …….” this sentence need reference confirming these genera are animal and human pathogen (many others are same). Consequently, I also recommended that authors design a table including plant, animal, and human pathogenic fungi with host and in the last column they can add the references.
- as we are as well unsure whether a merge of the results and discussion sections is allowed, we have not changed this.
Regarding Materials and methods,
what is the sampling technique used in this work?
-we included the sentence:
“The Gardenia fruits were collected as whole fruit, while the hard surface samples have been scratched with a scalpel and the plant surface samples have been cut from the plant.“
to the Methods section 2
Authors should write the GPS location of the study region and adding a satellite image for this area.
- we provide the GPS locations of the samples in the supplementary table S1. As the target samples (Gardenia) are from the city of Durban, we doubt that a satellite image may give the reader a better understanding of the habitat.
Also, authors need to explain why he choose this region; is it the only region in south Africa that has G. thunbergia. If it is not, authors should mention about the distribution of this ornamented plant in the south Africa in Introduction part.
- we now included a description about Gardenia thunbergia reading:
“G. thunbergia, commonly called the white or forest gardenia, is an evergreen shrub or small tree usually growing up to five metres tall with strongly scented white flowers 29. The hard woody indehiscent fruits are consumed by large mammals thus distributing the seeds. The species occurs naturally in South Africa (Eastern Cape and KwaZulu-Natal Provinces), Southern Mozambique, and eastern eSwatini (Swaziland).“
Finally, I would like to mention that all tables and figures in supplementary data are not appeared except table 2
- we are very sorry about this formatting issue. We hope that we addressed this now.
Best regards
Submission Date
13 April 2021
Date of this review
17 Apr 2021 03:12:57
Reviewer 2 Report
This manuscript characterized the microbe communities on the surface of the garden ornamental Gardenia thunbergia Thunb. fruit and compared them with the surrounding environmental microbe communities. The authors found several clades of fungi on the fruit surface may be the source of pathogenic fungi to plants and animals. Overall, the manuscript provided enough data, but they are not well organized. There are many technical issues that should be clarified and they are listed below.
- A major problem throughout the manuscript is lack of explanation of some key abbreviations. It will make non-expertise readers difficult to understand the whole logic. For example, in the introduction section, the full name of “ITS2” should be added when mentioned for the first time. In legend of Figure 1, the sample names “Afr_1” made people confused. I guess “Afr” may stand for “Africa”, but I still suggest to explain it in legend. In line 79, the meaning of “ASV” should be added too.
- The legend of Figure 1 is not sufficient to explain the figure. Please point out what the samples exactly are in the legend.
- Figure 4 is missing in the manuscript, only the legend of figure 4 presents in the manuscript. Meanwhile, I couldn’t see the figures in supplemental file. It may be caused by the format of figures the authors inserted, please carefully reorganize the manuscript.
- A mixed style of references is used, please carefully check them. For example, ref 39, ref 41, ref 43.
Author Response
Author's Reply to the Review Report (Reviewer 2)
Dear reviewer, unfortunately, we overlooked one ASV of a putative pathogen (ASV_003838, Cosmospora) previously. Hence, we needed to adjust the numbers in the text accordingly and included this sequence in Table1. Further, some minor changes in the results section (lines 225-228) and in the discussion (338-341 and 379-385) were necessary, as this led to an other interpretation of another sequence (ASV_004044) as well (being considered as a lichen fungus). In the whole ms, we omitted some minor typos and removed double spaces. All these corrections are marked with track changes.
Comments and Suggestions for Authors
This manuscript characterized the microbe communities on the surface of the garden ornamental Gardenia thunbergia Thunb. fruit and compared them with the surrounding environmental microbe communities. The authors found several clades of fungi on the fruit surface may be the source of pathogenic fungi to plants and animals. Overall, the manuscript provided enough data, but they are not well organized. There are many technical issues that should be clarified and they are listed below.
A major problem throughout the manuscript is lack of explanation of some key abbreviations. It will make non-expertise readers difficult to understand the whole logic. For example, in the introduction section, the full name of “ITS2” should be added when mentioned for the first time.
- as we used ITS2 now in the abstract, we have not written the full term in the introduction again.
In legend of Figure 1, the sample names “Afr_1” made people confused. I guess “Afr” may stand for “Africa”, but I still suggest to explain it in legend.
- we now included this in the legend of Fig. 1 as follows:
“Samples from South Africa used in this study.“
In line 79, the meaning of “ASV” should be added too.
- we now included the term ASV in the abstract and hence have not written the full term in the introduction again.
The legend of Figure 1 is not sufficient to explain the figure. Please point out what the samples exactly are in the legend.
- we added this. The whole legend now reads:
“Fig. 1: Samples from South Africa used in this study. Samples Afr_1 toAfr_9 (pictures 1 to 9) are from different surfaces of plants and artificial substrates and are used to characterize the microbial communities of the whole study site, while samples Afr_10 toAfr_16 are Gardenia thunbergia fruits (see Table S1).“
Figure 4 is missing in the manuscript, only the legend of figure 4 presents in the manuscript. Meanwhile, I couldn’t see the figures in supplemental file. It may be caused by the format of figures the authors inserted, please carefully reorganize the manuscript.
- we are very sorry about this formatting issue. Due to its relatively large size the figure was obviously not transferred. As the figures will be sent separately, we believe that this issue is now solved
A mixed style of references is used, please carefully check them. For example, ref 39, ref 41, ref 43.
- we used this to clearly seperate these from the previous text, especially in cases of parentheses or numbers, as otherwise the reference number may be interpreted as part of a formula.
Submission Date
13 April 2021
Date of this review
14 Apr 2021 19:57:27
Round 2
Reviewer 1 Report
Dear authors,
The manuscript has been improved with some good slight modification from the old version and it is good enough to be published in the present form
regards